# Highly tensile-strained Ge/InAlAs nanocomposites

Daehwan Jung[1], Joseph Faucher[1], Samik Mukherjee[2], Austin Akey[3], Daniel J. Ironside[4], Matthew Cabral[5], Xiahan Sang[5], James Lebeau[5], Seth R. Bank[4], Tonio Buonassisi[3], Oussama Moutanabbir[2] & Minjoo Larry Lee[1,6]

Self-assembled nanocomposites have been extensively investigated due to the novel properties that can emerge when multiple material phases are combined. Growth of epitaxial nanocomposites using lattice-mismatched constituents also enables strain-engineering, which can be used to further enhance material properties. Here, we report self-assembled growth of highly tensile-strained $Ge/In_{0.52}Al_{0.48}As$ (InAlAs) nanocomposites by using spontaneous phase separation. Transmission electron microscopy shows a high density of single-crystalline germanium nanostructures coherently embedded in InAlAs without extended defects, and Raman spectroscopy reveals a 3.8% biaxial tensile strain in the germanium nanostructures. We also show that the strain in the germanium nanostructures can be tuned to 5.3% by altering the lattice constant of the matrix material, illustrating the versatility of epitaxial nanocomposites for strain engineering. Photoluminescence and electroluminescence results are then discussed to illustrate the potential for realizing devices based on this nanocomposite material.

[1] Department of Electrical Engineering, Yale University, New Haven, Connecticut 06511, USA. [2] Department of Engineering Physics, École Polytechnique de Montreal, Montreal, Quebec, Canada H3C 3A7. [3] Department of Mechanical Engineering, Massachusetts Institute of Technology, Cambridge, Massachusetts 02139, USA. [4] Department of Electrical and Computer Engineering, University of Texas at Austin, Austin, Texas 78758, USA. [5] Department of Materials Science and Engineering, North Carolina State University, Raleigh, North Carolina 27606, USA. [6] Department of Electrical and Computer Engineering, University of Illinois at Urbana-Champaign, Urbana, Illinois 61801, USA. Correspondence and requests for materials should be addressed to D.J. (email: daehwan.jung@engineering.ucsb.edu) or to M.L.L. (email: mllee@illinois.edu).

Germanium (Ge) is a semiconductor material with a long history of research behind it, including the first transistor demonstration[1]. One of the most intriguing aspects of Ge is that strain can dramatically alter its band structure and enhance its optical and electrical properties. Spurred by theoretical work showing that tensile strain can convert Ge from an indirect-gap to a direct-gap semiconductor[2], recent research has focused on applying large biaxial or uniaxial tension using a range of approaches. For example, epitaxial growth of Ge thin films on template layers with larger lattice constant (for example, InGaAs or GeSn) has enabled biaxial tensile strain up to 2.33% (refs 3,4). Top-down fabrication techniques have also been used to fabricate Ge in biaxial or uniaxial tension using structures such as nanomembranes[5], bridges[6,7] and suspended nanowires[8,9].

Self-assembled nanocomposites grown by spontaneous phase separation have been investigated in diverse materials, including oxide compounds[10] and rare-earth monopnictide/III–V systems[11,12]. Furthermore, oxide nanocomposites with different lattice parameters enabling strain tuning and enhanced properties have been reported[13,14]. For semiconductors, only unstrained Ge/GaAs nanocomposites have been reported to date[15].

Here, we demonstrate spontaneous phase separation during molecular beam epitaxy (MBE) growth as an alternative approach to forming highly tensile-strained Ge nanostructures, coherently embedded in an InAlAs matrix (that is, Ge/InAlAs nano-composites). While the mutual immiscibility of Ge with III–V materials provides the driving force for phase separation[16], changes in growth kinetics enable significant control over nanostructure morphology, from nanowires to nanosheets. The tensile strain in the Ge nanostructures is confirmed by Raman spectroscopy, and we further show that changing the $In_xAl_{1-x}As$ matrix composition enables strain tuning up to 5.3%, which, to our knowledge, is the largest biaxial tension realized in Ge to date; the cross-over from indirect to direct is predicted at ∼2% biaxial tension[17]. We believe that the group-IV/III–V nanocomposites demonstrated here constitute a new materials platform for investigation of basic aspects of phase-separated growth, as well as offering the ability to create ultra-high strain states and properties that are otherwise inaccessible through conventional growth and processing.

## Results

**Structural properties of Ge/InAlAs nanocomposite.** Figure 1a schematically illustrates the Ge/InAlAs nanocomposite growth process, which was performed by codeposition of Ge, In, Al and $As_2$ over a range of growth temperatures, growth rates and Ge fluxes (see 'Methods' section). Atom probe tomography (APT) of a Ge/InAlAs nanocomposite layer with 3.6% Ge grown at 500 °C reveals vertically continuous, Ge-rich columns in the [001] growth direction (Fig. 1b). The cross-sectional annular dark-field (ADF) scanning transmission electron microscopy (STEM) and corresponding energy dispersive X-ray spectroscopy (EDX) Ge elemental mapping illustrate that phase separation occurs immediately after the Ge atoms are incorporated into the film without a two-dimensional wetting layer (Fig. 1c); a planar wetting layer is typically present in Stranski Krastanov growth mode of compressively strained nanostructures[18]. Planar-view images show that dark spots in the DF-STEM image precisely match the location of high Ge X-ray counts (Fig. 1d, see Supplementary Fig. 1 for EDX maps of Ge, In, Al and As). The diameter of the nanowires is ∼5.5 nm with a density of $1.8 \times 10^{10}$ cm$^{-2}$ in this sample. The nanowire densities can be

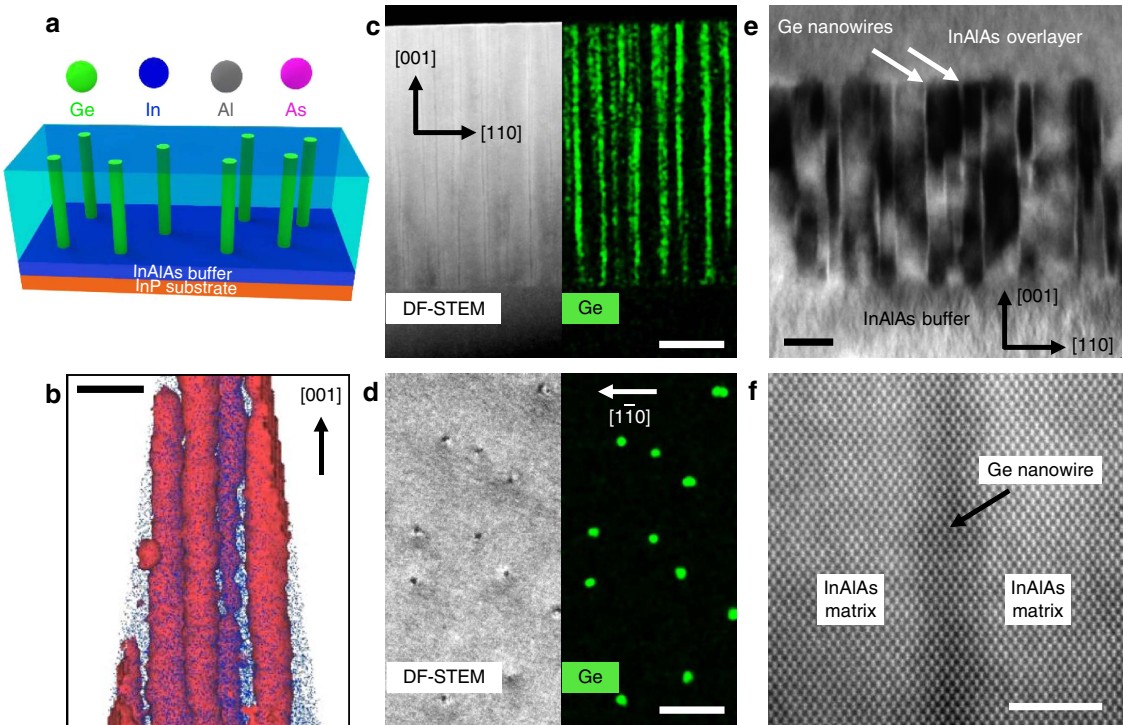

**Figure 1 | Structural characterization of Ge/InAlAs nanocomposite.** (**a**) Schematic illustration of Ge/InAlAs nanocomposite growth. (**b**) Three-dimensional APT image of Ge nanowires showing 10% Ge concentration iso-surface (red) and Al atoms (blue). Scale bar, 20 nm. (**c**) Cross-sectional ADF-STEM and EDX images showing continuous columnar growth of Ge nanowires. Scale bar, 80 nm. (**d**) Planar-view ADF-STEM and EDX Ge elemental mapping images. Scale bar, 60 nm. (**e**) Bright-field TEM image under $g = (220)$ two-beam condition shows no evidence of extended defects in a Ge/InAlAs nanocomposite layer sandwiched by InAlAs layers. Scale bar, 50 nm. (**f**) Cross-sectional $C_s$-corrected low-angle ADF-STEM image reveals single-crystalline Ge nanowires with fully coherent interfaces. Scale bar, 4 nm. Images **b,c** were taken from samples with 3.6% Ge, growth temperature = 500 °C and images **d–f** were taken from a sample with 1.4% Ge, growth temperature = 520 °C.

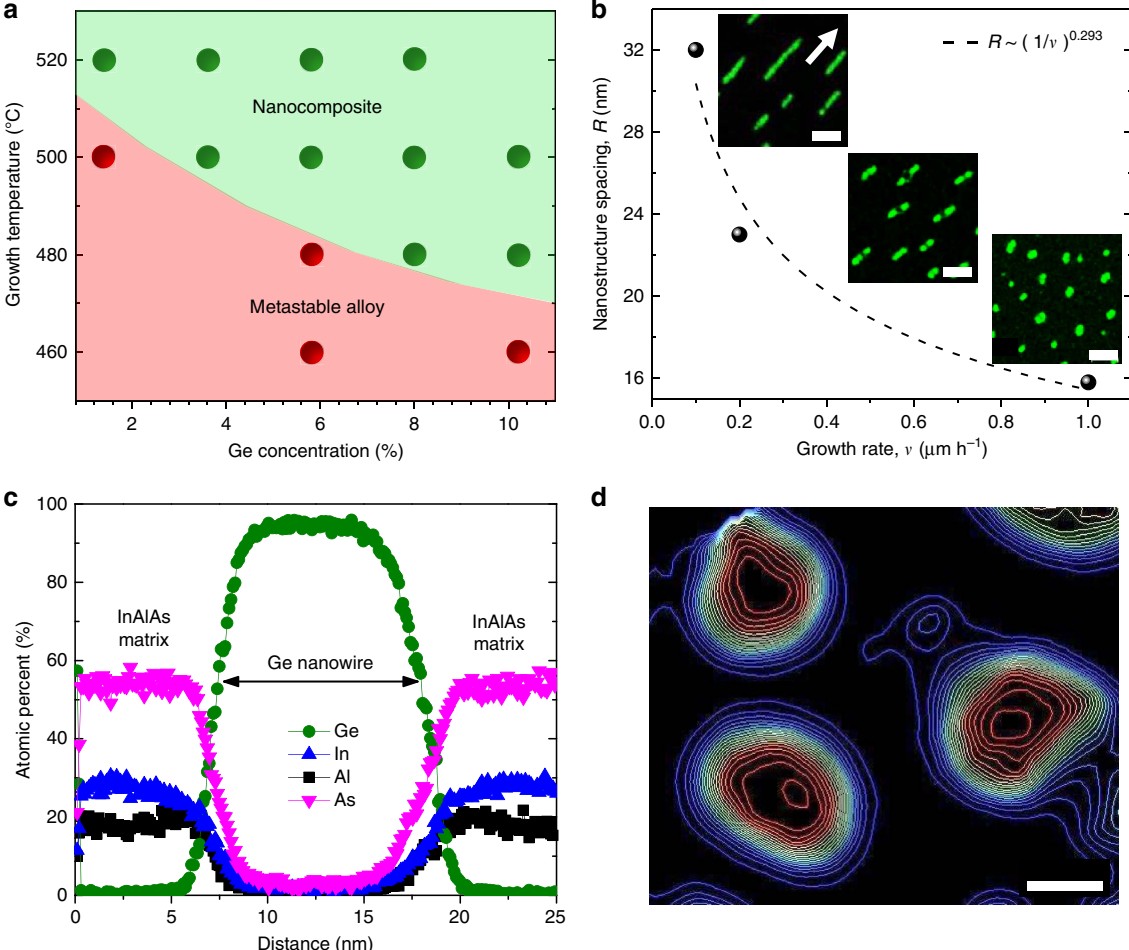

**Figure 2 | Effect of growth parameters.** (**a**) Nanocomposite growth diagram showing effect of growth temperature and %Ge on phase separation. Red circle denotes no observation of phase separation (Ge/InAlAs metastable alloy) and green circle denotes phase-separated growth (Ge/InAlAs nanocomposite). All samples were grown at 1 μm per hour. (**b**) Measured nanostructure spacings versus growth rate. Black dash curve is a power law fit. Insets show planar-view EDX Ge maps from 3.6% Ge samples grown at 1 μm per hour, 0.2 μm per hour, and 0.1 μm per hour growth rate. Scale bars are 30 nm and arrow direction is [1$\bar{1}$0]. (**c**) EDX linescans of Al, Ge, In, and As across nanowires. (**d**) APT Ge concentration iso-contour transect for nanowires. Outermost iso-contour line (blue) represents 5% Ge and innermost iso-contour line (red) is 95% Ge for each nanowire. Scale bar, 10 nm.

tuned by altering the Ge content in the nanocomposites (Supplementary Fig. 2 and Supplementary Note 1).

The bright-field TEM image in Fig. 1e shows coherent Ge nanowires without plastic strain relaxation, even with the high lattice mismatch of 3.72%. Furthermore, no extended defects such as stacking faults, anti-phase domains or dislocations are observed in either the nanocomposite layers or the Ge-free InAlAs overlayers. In contrast, nanocomposites with much higher Ge content did show plastic relaxation through dislocation formation due to the higher strain energy (Supplementary Fig. 3). Intense strain-induced contrast can be seen in the InAlAs matrix surrounding the nanowires, which will be discussed more in a later section. The aberration-corrected low-angle ADF-STEM image (Fig. 1f) shows a single-crystalline Ge nanowire and coherent boundaries between the nanowire and InAlAs matrix material with atomic resolution.

**Growth kinetics of Ge/InAlAs nanocomposite**. Unlike bulk phase separation at thermal equilibrium, spontaneous phase separation during MBE growth is surface-mediated, and therefore adatom kinetics strongly affect the final microstructures.

The diagram in Fig. 2a illustrates the basic growth kinetics for this materials system, with a Ge/InAlAs metastable alloy regime and a phase-separated nanocomposite regime; all samples were grown at 1 μm per hour. We find that higher growth temperatures are required for lower Ge content films to form a phase-separated structure; the fact that phase separation occurs at all for samples with 1.4% Ge (upper left of diagram) is testament to the low mutual solubility between Ge and InAlAs at 520 °C (Supplementary Fig. 4). Meanwhile, the lack of phase separation in 5.8–10.2% Ge samples grown at 460 °C shows that suppressing surface diffusion 'turns off' phase separation, resulting in metastable, single-phase layers (Supplementary Fig. 4). *Ex situ* annealing of the metastable Ge/InAlAs samples (red region of diagram) at 520 °C for 2 hours did not lead to any observable structural changes by TEM, confirming that the formation of phase-separated Ge nanostructures relies on surface diffusion[12,19], as opposed to bulk diffusion[20,21].

A decrease in growth rate leads to larger nanostructures with a lower density, consistent with previously reported experimental results on other nanocomposite material systems (Fig. 2b)[22]. The insets of Fig. 2b reveal that we can tune the morphology of the Ge nanostructures from nanowires to nanosheets by changing

**a**

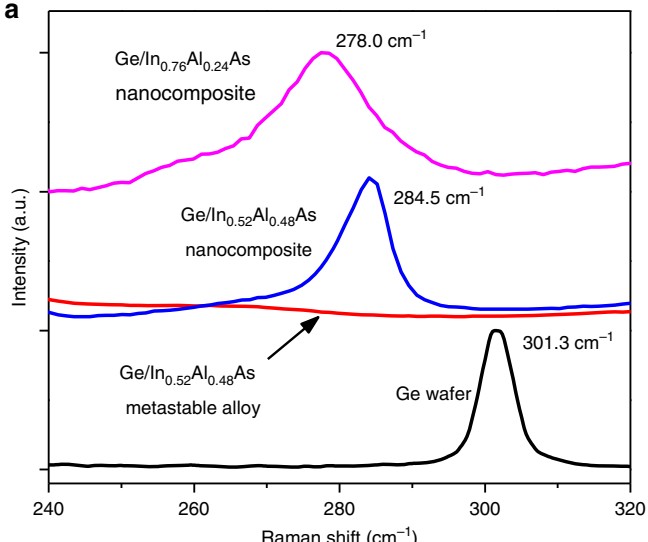

**b**

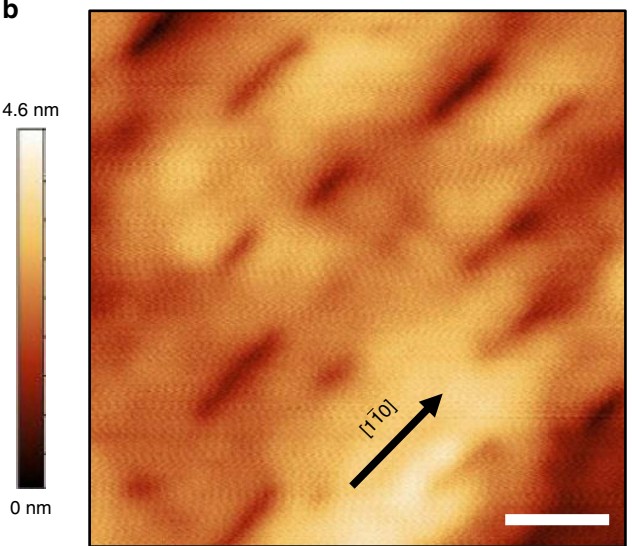

**Figure 3 | Raman spectroscopy and AFM image.** (**a**) Raman spectra of Ge wafer (black), Ge/In$_{0.52}$Al$_{0.48}$As metastable alloy (red), Ge/In$_{0.52}$Al$_{0.48}$As nanocomposite (blue), and Ge/In$_{0.76}$Al$_{0.24}$As nanocomposite (magenta) taken via 633 nm laser. Clear Raman shifts of Ge-Ge peaks are observed, showing tensile strain in the Ge nanostructures. (**b**) $250 \times 250$ nm$^2$ AFM image of Ge nanosheet sample showing surface depressions corresponding to the size and density observed in TEM. Scale bar, 50 nm.

the growth rate. The surface diffusion distance, $\rho = \sqrt{D_s \delta / \upsilon}$, is known to affect the size of the phase-separated nanostructures, where $D_s$ is the surface diffusivity, $\delta$ is the monolayer thickness and $\upsilon$ is growth rate[23]. As a result, the size of the Ge nanostructures increases with decreased growth rate, and the average spacing ($R$) between nanostructures is increased. Taking $R$ to represent the minimum surface diffusion distance of Ge adatoms during growth, we found that $\rho$ is proportional to $(1/\upsilon)^n$ with an exponent of 0.293. Our experimentally measured exponent is smaller than the theoretical value of 0.5 and the Monte Carlo simulation result of 0.39 by Adams et al.[23], but very close to the calculated exponent of 0.27 by He et al.[24]. Thus, despite the novelty of Ge/InAlAs nanocomposites, classic descriptions of phase separation can be used to predict and control basic aspects of their growth.

EDX linescans in Fig. 2c reveal that the Ge concentration of nanowires grown at 0.2 µm per hour peaks at ~95% while the Ge

concentration is abruptly diminished to <1% in the matrix. The APT transect in Fig. 2d further confirms that the centre of each nanowire is ≥95% Ge and that the Ge concentration in the surrounding matrix is <5% Ge (see Supplementary Fig. 5 for APT transects of In, Al and As). In contrast, the Ge concentration of nanowires grown at 1 µm per hour peaks at ~75% (Supplementary Fig. 6 and Supplementary Note 1). Thus, while the growth diagram in Fig. 2a shows that phase separation and nanocomposite formation occur readily, EDX and APT demonstrate that growth conditions encouraging long surface diffusion distance (for example, low growth rate or high growth temperature) increase the extent to which phase separation takes place.

**Strain properties of Ge/InAlAs nanocomposite.** Raman spectroscopy in Fig. 3a shows that a 3.6% Ge/InAlAs nanocomposite sample (blue) has a Raman peak at 284.5 cm$^{-1}$, which is strongly shifted from the bulk Ge (black) longitudinal optical (LO) phonon peak at 301.3 cm$^{-1}$. Care was taken to minimize any effects from sample heating, and while phonon confinement may contribute to the observed shift[25], biaxial tensile strain is the dominant cause (Supplementary Figs 7 and 8 and Supplementary Note 2). We deduced the degree of strain in the Ge nanostructures using the equation, $\Delta\omega = b \cdot$ biaxial tension, where $b$ is a phonon strain shift coefficient; a value of $b = -440$ cm$^{-1}$ is adopted here[26]. The calculated tensile strain is 3.8%, which is extremely close to the lattice mismatch between Ge and InAlAs (3.72%), and consistent with the Ge nanostructures being under biaxial tension near the surface. Surface depressions in the atomic force microscopy (AFM) image (Fig. 3b) matching the shape and size of Ge nanostructures observed in TEM are also consistent with the out-of-plane elastic relaxation expected from biaxially tensile-strained structures at a free surface[27]. For comparison, the 3.6% Ge/InAlAs metastable alloy sample grown at 400 °C (red in Fig. 3a) shows only the In–As LO phonon peak at ~230 cm$^{-1}$ and Al–As LO phonon peak at ~360 cm$^{-1}$ (Supplementary Fig. 9), but no additional peak near 284.5 cm$^{-1}$. It is important to note that Raman spectroscopy analyzes only the top portion of the Ge nanostructures, because the penetration depth of the 633 nm laser is only ~30 nm in Ge. On the basis of the observed lack of dislocations at the Ge/InAlAs interfaces, we speculate that the Ge nanostructures may take on a triaxial strain state further beneath the surface, where elastic relaxation is not possible, but a detailed analysis of strain versus depth is left for future work.

A further shifted Ge LO peak (magenta in Fig. 3a) is observed from Ge nanostructures embedded in an In$_{0.76}$Al$_{0.24}$As matrix, which demonstrates the ability to control the strain in the Ge nanostructures (Supplementary Figs 10 and 11 and Supplementary Note 3 for more details regarding the InAlAs graded buffer). The observed 23.3 cm$^{-1}$ shift to 278.0 cm$^{-1}$ corresponds to a 5.3% biaxial tensile strain, which is again, very close to the lattice mismatch between the Ge and In$_{0.76}$Al$_{0.24}$As matrix (5.15%); for comparison, Raman shift values of 5.7–9.7 cm$^{-1}$ have been reported in earlier work on biaxial tensile-strained Ge[3,28]. Our results demonstrate that the tensile strain in phase-separated Ge nanostructures is readily tunable to much higher values by changing the matrix lattice constant.

**Raman polar patterns and optical properties.** The Ge nanosheet sample displays strongly polar Raman intensity patterns following $I_{Raman} \sim \cos^4(\theta)$ (Fig. 4a). Previous studies reported that Raman scattering in semiconductor nanowires with diameter of ~50 nm can be dominated by a nearly perfect optical antenna effect, which is strong enough to mask the well-established bulk Raman

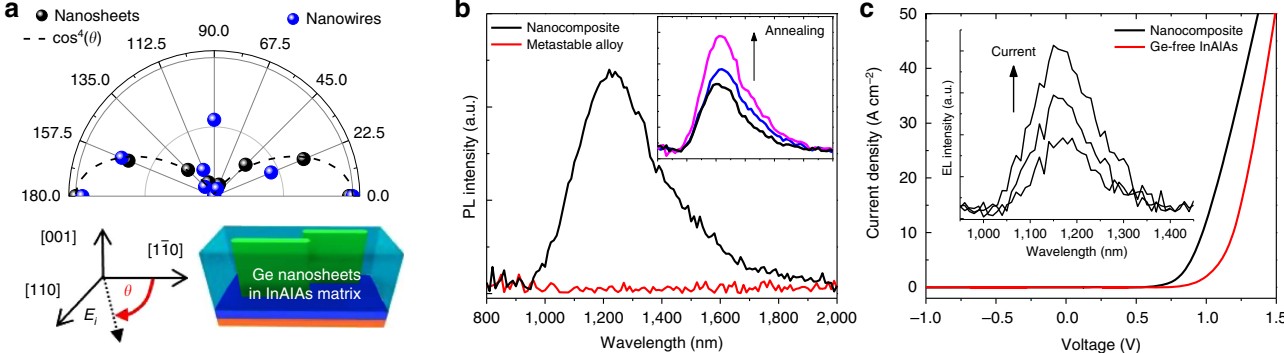

**Figure 4 | Raman scattering and luminescence.** (**a**) Polar ($\theta$) plots of Ge LO Raman peaks from Ge nanosheets (black circles) and nanowires (blue circles). The Raman intensities are normalized. Inset illustrates that $\theta$ is the angle between polarization of incident laser ($E_i$) and [1$\bar{1}$0] direction of the sample. (**b**) Room-temperature PL spectra from Ge/InAlAs nanocomposite (black) and from Ge/InAlAs alloy (red). Inset shows annealing effect of nanocomposite PL intensity at room-temperature. (**c**) J–V curves from nanocomposite p-n diode (black) and Ge-free InAlAs diode (red). Inset shows EL spectrum from the nanocomposite diode at room-temperature.

selection rules[29,30]. Because the Ge nanosheets possess strong structural anisotropy, the Raman intensity exhibits a clear dependence on the angle ($\theta$) between the incident laser polarization direction and the Ge nanosheet orientation. The fourth power dependence of Raman scattering intensity was observed previously in studies of single silicon nanowires, and explained by a resonant enhancement of both the initial excitation light (532 nm) and the Raman scattered light (Stokes shifted to 540 nm)[30]; such effects may also help to explain the fourth power dependence observed here. To the best of our knowledge, this is the first observation of polarization-dependent Raman scattering intensity from semiconductor nanocomposites. Nanocomposites with embedded Ge nanowires (blue circles) do not follow the $\cos^4(\theta)$ behaviour, and the effect of the bulk Raman selection rules are still present; the Ge nanowire diameter may be too small to exhibit strong optical antenna effects.

Figure 4b shows a broad photoluminescence (PL) signal peaked at ~1,230 nm from the 3.6% Ge/InAlAs nanocomposite at room temperature. In contrast, a Ge/InAlAs metastable alloy that contains the same 3.6% Ge has no PL response in this wavelength range, indicating that the ~1.0 eV PL peak is from the phase-separated Ge nanostructures; the bulk bandgap of InAlAs is ~1.53 eV. The theoretically calculated direct bandgap of 3.75% biaxial tensile-strained Ge is ~0.49 eV from Gamma ($\Gamma$) to heavy-hole (HH) and ~0.12 eV from $\Gamma$ to light-hole (LH)[17]. In our surface-emission configuration, only $\Gamma$–HH emission can be collected, because the light emitted from $\Gamma$–LH recombination propagates in the in-plane direction[5]. The large blue-shift from 0.49 to 1.0 eV likely results from strong quantum confinement effects and unintentional doping, while the high full-width at half-maximum indicates inhomogeneous broadening. However, the possible role of the non-uniform strain state of the nanostructures along with the different stoichiometry at the interfaces between Ge and InAlAs also needs to be investigated[31].

We also performed temperature-dependent PL and low-temperature ($T = 77$ K) excitation-dependent PL on the 3.6% Ge/InAlAs nanocomposite (Supplementary Fig. 12 and Supplementary Note 4). The integrated PL intensity increases and the PL peak blue-shifts as the temperature decreases, which is consistent with direct bandgap radiative recombination[3,32]. Excitation-dependent PL revealed that the PL intensities vary linearly with the excitation density. The inset of Fig. 4b shows that annealing up to 600 °C can improve the PL intensity by ~2×, which may result from a reduced density of point defects in and around the Ge nanostructures[33]. Further study of optical properties such as time-resolved PL[31] and photoacoustic

spectroscopy[34] will be necessary to confirm the band structure of the highly tensile-strained Ge/InAlAs nanocomposite.

We grew and fabricated light-emitting diodes (LEDs) using a Ge/InAlAs nanocomposite active region to investigate its potential for device applications. The I–V curve in Fig. 4c shows that both the nanocomposite and control device act as well-behaved diodes with good rectification and low leakage current, showing that the Ge nanostructures do not serve as significant shunt paths. Extracted ideality factors for the Ge/InAlAs nanocomposite LEDs and InAlAs diodes were ~2 due to a mixture of recombination in the depletion and quasi-neutral regions. The inset of Fig. 4c shows an electro-luminescence (EL) peak at $\lambda = 1,170$ nm at room temperature, about 50 meV blue-shifted from the PL emission peak. The integrated EL intensity increases approximately linearly as the current density increases from 22 to 46 A cm$^{-2}$. These initial device results demonstrate the possibility of integrating highly tensile-strained Ge/InAlAs nanocomposites into p–n junction devices such as light emitters and detectors.

## Discussion

In summary, we demonstrated the use of surface-mediated phase separation to realize highly tensile-strained Ge/InAlAs nanocomposites. TEM showed coherently strained, single-crystalline Ge/InAlAs nanocomposites without observable extended defects, while Raman spectroscopy revealed 5.3% biaxial tensile strain, to our knowledge the highest value reported to date in Ge nanostructures. Ge/InAlAs nanocomposites are a new class of material allowing study of the electronic and optical properties of tensile-strained Ge, and the growth method may be extendable to other strained, semiconductor-based nanostructures for future optoelectronic applications.

## Methods

**Molecular beam epitaxy growth.** Growth was conducted using a Veeco Modular Gen-II MBE system. A 200 nm lattice-matched InAlAs buffer was first grown on (001) InP at 500 °C. Next, 300 nm nanocomposite layers consisting of tensile Ge nanostructures embedded in InAlAs were grown by codeposition of Ge, In, Al and As$_2$ over a range of substrate temperatures, growth rates and Ge fluxes. The V/III ratio for the Ge/InAlAs film was kept at ~25, and reflection high-energy electron diffraction patterns were streaky throughout the growth.

**Structural characterization via transmission electron microscopy.** TEM samples were thinned by standard mechanical grinding and additional Ar ion-milling. ADF-STEM (camera length 220 mm) and EDX elemental mapping and linescans were performed on a Tecnai Osiris microscope at an operating voltage of 200 kV. Low-angle ADF-STEM was performed using a probe corrected

FEI Titan G2 60–300 kV operated at 200 kV with a beam current of $\sim 50$ pA. The collection inner semi-angle and probe semi-convergence angle were $\sim 28$ mrad and 21 mrad, respectively. Atomic resolution STEM images were acquired with the revolving STEM (RevSTEM) method to eliminate drift distortion[35]. Each RevSTEM data set contained eight image frames of size 2,048 × 2,048 pixels with a 90° rotation between each successive frame. The time-dependent drift vector was measured from the rotating frames, which were then corrected, aligned and averaged to yield the final RevSTEM image.

**Raman spectroscopy.** Backscattering micro-Raman spectroscopy was carried out in ambient air in a Renishaw InVia RM 3,000 set-up, using a He/Ne 633 nm continuous-wave laser. A 100× Olympus objective (spot size at laser waist $\sim 1.0 \mu m$) was used to focus the laser onto the samples, and the backscattered light was diffracted by an 1,800 lines per mm grating onto a liquid nitrogen-cooled CCD camera. The incident laser power density of $1.72$ mW $\mu m^{-2}$, as measured by a miniature power metre, was kept constant throughout the experiment. The data was collected from three different spots on each sample and averaged.

**Atom probe tomography.** APT specimens were prepared by standard focused ion beam (FIB) lift-out procedures. Using a Zeiss NVision 40 dual beam SEM/FIB (Ga$^+$) equipped with an Omniprobe micromanipulator, sections of the Ge/InAlAs nanocomposite were attached to prefabricated micro tip coupons. Individual needle-shaped APT specimens were then fabricated using FIB annular milling at an accelerating voltage of 30 keV, followed by final sharpening and cleanup with an accelerating voltage of 5 kV. Finished specimens had a tip radius-of-curvature of $\sim 30$ nm, and a shank half-angle of $\sim 18$ degrees, as measured by SEM. APT was performed using a Cameca LEAP 4,000 HR equipped with a 355 nm laser; specimens were run with a base temperature of 40 K, laser energy of 5 pJ per pulse, repetition rate of 100 kHz and target detection rate of 1%. At these conditions, significant As clustering was observed, as is common in III–V materials. APT data sets were then reconstructed and analysed using Cameca's integrated visualization and analysis software.

**Photoluminescence/electroluminescence.** Samples were optically pumped using a 532 nm laser (photon energy significantly greater than the bandgap of InAlAs matrix and InP substrate), and optical emission was collected by a spectrometer with a liquid nitrogen-cooled InSb detector. The LEDs were tested by DC current injection and the light was collected by an InGaAs detector cooled to 253 K.

**Light-emitting diode fabrication.** We grew and fabricated p–n LED using a 3.6% Ge/InAlAs nanocomposite active layer (Supplementary Fig. 13a). We found that all of the nanocomposite layers are $1$–$3 \times 10^{18}$ cm$^{-3}$ n-type using room-temperature Hall effect measurements. Be-doped InAlAs was grown on top of the nano-composite layers to create p-n junction diodes. The LEDs were fabricated by defining $1 \times 1$ mm$^2$ mesas, and gridded front metal contacts (Ti/Au = 20 nm/ 200 nm) and planar back metal contacts (AuGe/Ni/Au = 100 nm/20 nm/100 nm) were deposited for the anode and cathode, respectively. After front metal lift-off and mesa definition, the p-type InGaAs contact layer was etched using citric acid and hydrogen peroxide to improve light extraction. Devices were tested under DC current injection at room-temperature, and the light was collected by an InGaAs detector cooled to 253 K. Supplementary Fig. 13b shows J–V curves from the p-n nanocomposite LED and Ge-free InAlAs p–n junction in a semi-log plot.

**Data availability.** The data that support the findings of this study are available from the corresponding authors on request.

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

## Acknowledgements

This work was supported by NSF DMR 1506371 as well as a Dubinsky New Research Initiative Grant. Facilities use was supported by YINQE and NSF MRSEC DMR 1119826. The work at École Polytechnique was supported by NSERC (Discovery Grant), Canada Research Chair, and Canada Foundation for Innovation. The work at MIT was supported

Bay Area Photovoltaic Consortium (BAPVC) under Contract No. DE-EE0004946. The work at UT-Austin was supported by a Multidisciplinary University Research Initiative from the Air Force Office of Scientific Research (AFOSR MURI Award No. FA9550-12-1-0488) and NSF DMR 1508603. The authors acknowledge the use of the Analytical Instrumentation Facility (AIF) at North Carolina State University, which is supported by the State of North Carolina and the National Science Foundation.

## Author contributions

D.J., J.F., and M.L.L. conceived the idea, and D.J. and M.L.L. designed the experiments. D.J. grew the nanocomposite materials and performed DF-STEM, EDX, TEM, XRD, AFM, and Raman spectroscopy via 532 nm laser. D.J. fabricated the LEDs and measured I-Vs. A.A. and T.B. conducted the APT on the nanocomposites. S.M. and O.M. performed Raman spectroscopy via 633 nm laser. M.C., X.S. and J.L. carried out $C_s$-corrected low-angle ADF-STEM. D.J.I. and S.R.B. conducted PL and EL measurements. D.J. and M.L.L. wrote the paper with contributions from all the other authors.

## Additional information

**Competing financial interests:** The authors declare no competing financial interests.

**Publisher's note**: 

