## [Peer Review File · Nature Communications]

Reviewers' comments:

Reviewer #1 (Remarks to the Author):

This is a creative and technically sound study demonstrating that strain (and thus electronic behavior) can be tailored in nanocomposites grown in immiscible systems. It is thorough in that the materials are well-characterized, a "map" of appropriate processing conditions are presented, and theoretically grounded. Two minor suggestions that might clarify some of the authors' explanations:

1. The authors can strengthen the manuscript by elaborating on features of the data which they use to draw conclusions. For example, what features of Fig. 1e demonstrate the lack of plastic strain relaxation? It would be helpful for readers to point features here, and elsewhere in the data interpretation.
2. Similarly, a somewhat better description of Fig. 2a would be helpful, such as an introductory/background sentence.

Reviewer #2 (Remarks to the Author):

The paper submitted by D. Jung et al to Nature Communications addresses a very popular research topic: namely strain induction and strain engineering in nanostructures. The results presented by the authors suggest up to 5.3% biaxial strain in Ge through the formation of Ge/InAlAs nanocomposite. Even though the value authors achieved for strain is encouraging and may start initiative to implement these nanocomposite structures into device architectures, I do not think the current manuscript meet the standards of the journal in terms of novelty and conclusive results.

(1) Firstly, phase separation during growth of nanostructures through self-assembly mechanism to produce ordered arrays of nanowires are well documented for both group IV and III-V systems. This article does not add towards this growth technique. Authors should clearly clarify and explain the novelty of their growth with respect to the published data on phase separation growth. It will also be better if the authors clearly demonstrate tuneability of the morphology (e.g. diameter) of the nanocomposite to show the diversity and range of their growth technique.

(2) Authors have calculated large biaxial strain of 5.3% from Raman spectroscopy. They have observed a massive shift of Raman peak (23.3 cm^{-1}) bulk Ge LO phonon mode. But they have not considered diameter dependence (or lack of it) on the Raman spectra of Ge nanostructures. For nanowire of 5.5 nm dimension Raman peak shift can also occur due to phonon confinement and relaxation of the $q \approx 0$ selection rule. Authors need to compare the position of their Raman peak with the Raman spectra from unstrained Ge nanowire of similar dimension to have conclusion on the observed Raman shift.

(3) Authors also need to justify the lack of dislocation at the interface between nanowire and the matrix for their growth. Is this due to particular geometry of the nanocomposite or the growth technique?

(4) Main motivation for strain induction in Ge nanowire is to get a direct bandgap transition from indirect semiconductors like Ge. But the assertion that these materials exhibit a direct band gap is not conclusive in the manuscript and mainly based on assumption. The optical data are in my opinion preliminary and require further study to confirm the direct bandgap and quantum confinement. Authors should demonstrate temperature dependent photoluminescence (at low temperatures) and measure the radiative rates of the transition to have more insight on the bandgap of this material.

(5) Also there are important missing references. For example, in the introduction, there is no reference to back up the statement "the cross-over from indirect to direct is predicted at 2% biaxial tension".

Reviewer #3 (Remarks to the Author):

The article reports a novel procedure for the fabrication of [100] oriented Ge nanowires in InAlAs planar layers by phase segregation. The interest in tensile strained Ge stems from the possibility of creating light emitters and detectors on a Si platform. While this work deviates from the popular attempts of co-integration of tensile stressed Ge on Si, there is indeed technological merit in realizing direct gap Ge on InP, which is used in high-speed base stations and cell phones. The addition of Ge can possibly bring low power optical modulators, light emitters and photodetectors as we move toward high bandwidth high bit rate optoelectronic transceivers. This or what the authors believe is of technological merit for the demonstrated growth should be highlighted in the manuscript introduction. Undoubtedly, the creation of defect free [100] Ge nanowires is important from a scientific perspective, as the popular techniques to realize quantum wires usually utilize [111] orientation that is seeded by supersaturated foreign metal eutectics; which are usually contaminated and sometimes defective. The ability to create small diameter nanowires to comply to high levels of strain enabled defect-free tensile strained Ge nanowires which resulted in direct bandgap controlled PL and EL spectra. The study is systematic, comprehensive, and novel, and I do believe that it should be published in Nature Communications.

There are a few aspects that I'm not clear about and that the authors should clarify in the manuscript on its way to publication:

1. In the best-case scenario, the Ge composition is 95%, as seems to be the highest composition 'alloyed' material from the EDS. Al is shown in APT, why not show side-by-side Ge, As, Al, In? There is possibly a great deal of doping compensation going on. With dominant As, an n-type carrier density in excess of $1E18/cm^3$ is reported. This of course would not fold back to the incorporated As in Ge from EDS which shows that As is actually a few % of the composition. This leads to the question of whether one should mention Ge nanowire at all or just discuss Ge/InAlAs composite everywhere on the manuscript? What is the lattice incorporation mechanism, are there any predicted crystal structures or can first principles predict a stable alloy or compound structure?
2. What is the nature of the band-offsets in the InAlAs/Ge system? Any possible rectification effects or is the conduction band-offset negligible?
3. LED devices: 'well-behaved diodes with good rectification and low leakage current'. Why not show currents on a log scale for leakage current visualization at least in supporting information.
4. Some discussion on tensile stress in [100] orientation and its effects on different conduction band-edge minima would be helpful

P.S. I really like your results. I really don't like the way you presented your figures. It gave me hard time flipping back and forth to know what panel is for which figure and then what caption is for which panel. Why not make the damn figure with its caption on one page and incorporate it in the text of the manuscript.

Reviewer #1

We are grateful for the comment that our paper shows “**a creative and technically sound study demonstrating that strain (and thus electronic behavior) can be tailored in nanocomposites grown in immiscible systems.**”

Comment #1. The authors can strengthen the manuscript by elaborating on features of the data which they use to draw conclusions. For example, what features of Fig. 1e demonstrate the lack of plastic strain relaxation? It would be helpful for readers to point features here, and elsewhere in the data interpretation.

Response: The lack of dislocations or other extended defects above and within the nanocomposite layer of Fig. 1e shows the lack of plastic strain relaxation; dislocation activity is synonymous with plasticity. To further clarify this result, we have added TEM data to the Supplementary Information from another sample where dislocations did form in the nanocomposite layer due to excessively high Ge content (see response to Reviewer #2, comment 3 below). We also added the following text to the Supplementary Information section:

“The cross-sectional $g = (220)$ BF-TEM image in Fig. S8a shows strain-induced contrast surrounding the Ge nanowires and the lack of plastic strain relaxation at the interfaces”.

Comment #2. Similarly, a somewhat better description of Fig. 2a would be helpful, such as an introductory/background sentence.

Response: We agree with the reviewer’s comment, and we added an introduction sentence for Fig. 2a at the beginning of the *Growth kinetics of Ge/InAlAs nanocomposite* section of the manuscript.

“Unlike bulk phase separation at thermal equilibrium, spontaneous phase separation during MBE growth is surface-mediated, and therefore adatom kinetics strongly affect the final microstructures.”.

Reviewer #2

The reviewer raised several concerns in terms of novelty and conclusive results. The concerns from the reviewer are reproduced below, and we have addressed them in a point-by-point structure.

Comment #1. Firstly, phase separation during growth of nanostructures through self-assembly mechanism to produce ordered arrays of nanowires are well documented for both group IV and III-V systems. This article does not add towards this growth technique. Authors should clearly clarify and explain the novelty of their growth with respect to the published data on phase separation growth. It will also be better if the authors clearly demonstrate tunability of the morphology (e.g. diameter) of the nanocomposite to show the diversity and range of their growth technique.

Response: We agree that self-assembled growth of ordered arrays of *free-standing* nanowires is well-established now. However, we believe that our work is distinctive and unique from that body of literature, since our technique allows formation of nanowires *embedded in a matrix, where altering the matrix allows*

tuning of strain. If we then compare our work to previous reports on embedded nanostructures grown by phase separation, we not only show tensile strained Ge nanostructures that are coherently embedded in a III-V matrix for the first time, but also demonstrate strain tuning in the Ge nanostructures, up to 5.3 %. A few studies have been reported about phase-separated, unstrained Ge/GaAs structures, but none of them clearly revealed columnar growth of Ge nanostructures or demonstrated the possibility of tensile strained Ge nanostructures. Thus, we believe that our work is distinct from both earlier work on nanowire array growth and from earlier work on nanocomposite growth. To emphasize the novelty of our work, we added an additional introductory paragraph as below.

“Self-assembled nanocomposites grown by spontaneous phase separation have been investigated in diverse materials, including oxide compounds¹ and rare-earth monpnictide/III-V systems^{2, 3}. Furthermore, oxide nanocomposites with different lattice parameters enabling strain tuning and enhanced properties have been reported^{4, 5}. For semiconductors, only unstrained Ge/GaAs nanocomposites have been reported to date⁶”.

Also, the reviewer mentioned that we should clearly clarify tunability of the morphology of the nanocomposites to show the diversity and range of our growth technique. While we stated that “changes in growth kinetics enable significant control over nanostructure morphology, from nanowires to nanosheets” in the introduction, an additional sentence below is added to describe the morphology tunability of the Ge nanostructures in the *Growth kinetics of Ge/InAlAs nanocomposite* section of the manuscript.

“The insets of Fig. 2b reveal that we can tune the morphology of the Ge nanostructures from nanowires to nanosheets by changing the growth rate”.

Comment #2. Authors have calculated large biaxial strain of 5.3% from Raman spectroscopy. They have observed a massive shift of Raman peak (23.3 cm⁻¹) bulk Ge LO phonon mode. But they have not considered diameter dependence (or lack of it) on the Raman spectra of Ge nanostructures. For nanowire of 5.5 nm dimension Raman peak shift can also occur due to phonon confinement and relaxation of the q≈0 selection rule. Authors need to compare the position of their Raman peak with the Raman spectra from unstrained Ge nanowire of similar dimension to have conclusion on the observed Raman shift.

Response: In general, the red-shift in Raman spectrum of the Ge-Ge LO phonon mode can occur due to the following reasons: (1) temperature rise due to laser heating, (2) phonon confinement, and (3) strain. First, it is very important to note that heating is pronounced for nanowires that are suspended (not in contact with any substrate) or at high laser power⁷. In our work, we eliminated this effect by keeping the laser power in our experiments below 1 mW (power density of 1.72 mW/μm²). Moreover, the fact that the Ge nanowires are embedded in the InAlAs matrix means that the little heat that is generated can quickly dissipate away without affecting Raman spectra.

Second, phonon confinement does induce a small red-shift, but the magnitude of the shift is small compared to what we observed experimentally. The more pronounced effect of phonon confinement is the asymmetric broadening. In unstrained Ge nanowires of diameter 5 nm [See figure 7a of Ref. ⁸], the observed confinement-induced red shift is only about 3 cm⁻¹ compared to bulk germanium. Phonon confinement therefore contributes only a small fraction of the huge red-shift of about 20 cm⁻¹ observed in our experiments, which we attribute mainly to biaxial strain. It is also important to note that the effect of phonon confinement was reported for VLS nanowires dispersed on a substrate. **This is different from our case where nanowires are embedded in a crystalline matrix. The interface between the matrix and nanowires is coherent, and thus the two lattices are coupled allowing phonons to propagate from nanowires to matrix and vice versa.** To fully decouple the weak effects of phonon confinement in these nanocomposite materials, Raman measurements would need to be performed on Ge nanowires that are not strained yet embedded inside an InAlAs matrix, making the synthesis of such nanowires itself beyond the scope of the current work.

Additionally, since the Ge nanowires are embedded within the host lattice, phonon confinement may arise from mass periodicity (Ge/InAlAs) which leads to folding of the Brillouin zone, as was observed for GaAs/AlAs superlattice⁹ or even isotopic superlattices¹⁰. However, phonon confinement due to mass periodicity is only observable at very low temperatures and can also be eliminated from the room temperature measurements conducted on our samples.

In conclusion, based on the reasons above, we are fully confident that the observed strong red shift in our experiments is mainly due to biaxial tensile strain. We added the following statement in the *Strain properties of Ge/InAlAs nanocomposite* section of the manuscript:

“Care was taken to minimize any effects from sample heating, and while phonon confinement may contribute slightly to the observed shift⁸, biaxial tensile strain is the dominant cause (Supplementary Info)”.

We also added the following sentences in the Supplementary Information:

“We eliminated heating as a possible cause of the observed red-shift by performing the Raman measurements at low incident laser power, specifically for this purpose. Moreover, since the Ge nanowires are embedded in an InAlAs matrix, the little heat that might have generated can be quickly dissipated away. We rule out phonon confinement because, for unstrained freely dispersed Ge nanowires, the observed confinement-induced red shift is very small compared to what we observed in our experiments⁸. Furthermore, the interfaces between the InAlAs matrix and Ge nanowires are coherent, meaning that the two lattices are coupled and phonons can propagate from nanowires to matrix and vice versa, minimizing further the chances of phonon confinement. Phonon confinement due to Ge/InAlAs mass periodicity is also eliminated since confinement in such structures are only observable at very low temperatures^{9, 10}”.

Comment #3. Authors also need to justify the lack of dislocation at the interface between nanowire and the matrix for their growth. Is this due to particular geometry of the nanocomposite or the growth technique?

Response: We believe that it is primarily due to the geometry of the Ge/InAlAs nanocomposite, where a relatively small amount of strained material (~3.6% of the volume) is sparsely distributed throughout the matrix. The moderate growth temperature of ~500 °C also probably leads to relatively slow dislocation kinetics. We did observe formation of extended defects in samples with very high Ge content (Ge ~12.4%) as shown in Figure R1 below. The high Ge content and high density of nanowires increases the net tensile strain energy of the nanocomposite, and eventually the critical strain energy for dislocation formation is exceeded. We also speculate that higher growth temperatures (e.g. 600 – 700 °C) might lead to earlier onset of dislocation activity due to the exponential increase in dislocation nucleation rate with temperature.

Figure R1. Cross-sectional BF-TEM image of high Ge content (10.2% and 12.4%) Ge/InAlAs nanocomposites showing dislocations.

To address the reviewer's comment, we added the sentences below in the *Structural properties of Ge/InAlAs nanocomposite* section of the manuscript.

In contrast, nanocomposites with much higher Ge content did show plastic relaxation through dislocation formation due to the higher strain energy (Supplementary Info).

For the Supplementary Information, we added figure R1 above and the paragraph below.

“Figure 1(e) of the manuscript shows that nanocomposites with relatively low Ge content are free of dislocations and stacking faults, indicating a lack of plastic relaxation. However, for Ge/InAlAs nanocomposites with relatively high Ge content (~12.4%), we observed extended defects as seen in Fig S5”.

Comment #4. Main motivation for strain induction in Ge nanowire is to get a direct bandgap transition from indirect semiconductors like Ge. But the assertion that these materials exhibit a direct band gap is not conclusive in the manuscript and mainly based on assumption. The optical data are in my opinion preliminary and require further study to confirm the direct bandgap and quantum confinement. Authors should demonstrate temperature dependent photoluminescence (at low temperatures) and measure the radiative rates of the transition to have more insight on the bandgap of this material.

Response: This is a paper describing the growth, structure, and strain engineering of a novel material system, and the main emphasis is on these characteristics. More detailed spectroscopic studies will yield greater understanding, but are largely outside of the scope of this paper. Overall, we agree with the reviewer that, despite the high strain in the Ge nanostructures, it is difficult to prove that they are direct gap, given the possible impact of interfaces, unknown band alignment, and unintentional doping, which will affect the electron/hole collection and recombination, in turn affecting the thermal- and temporal-dependent dynamics that the reviewer suggests. For example, it is possible to have an indirect bandgap material with a short carrier lifetime due to a high density of deep level defects, or a direct bandgap material with a long carrier lifetime in the presence of shallower defects. It is also possible to have a material with bandgap in the infrared with PL emission in the visible, as was the case in InN. Absorption measurements could clarify the issue; however, significantly thicker layers than those grown here will be required to clearly resolve the absorption edge. Similarly, low-temperature PL would be ideal to look for phonon replica peaks, but this requires extremely low excitation densities, which will require further materials optimization to improve injection efficiency into the Ge and so forth to produce detectible luminescence at such low excitation densities. Future studies might focus on these in concert with photoacoustic spectroscopy, which is sensitive to both direct and indirect transitions (the sensitivity of photo- and electro-reflectance techniques suffers from the significant optical scattering in nanowire samples)¹¹.

Following the reviewer's excellent suggestion, we performed low-temperature ($T = 77$ K) excitation-dependent PL (Figure R2a) and found that the peak and integrated PL intensities vary linearly with the excitation density; a linear dependence of integrated PL intensity on pump power is extremely common in studies of novel optoelectronic materials, including InN¹² and InGaAsN¹³. The sub-quadratic relationship between PL intensity (I_{PL}) and excitation power (P_{EX}) may be due to exciton-dominated radiative recombination in the Ge nanostructures¹⁴ or loss of carriers to non-radiative trap states^{13, 15}, such as at the Ge and InAlAs interfaces¹⁶. Therefore, it is difficult to conclude whether or not the tensile-strained Ge nanostructures have a direct bandgap. Temperature-dependent PL, as shown in Figure R2b, displays an integrated PL intensity increase (Figure R2c) and a blue-shift of the PL peak as temperature decreases, both of which are common PL characteristics of direct bandgap materials. Note that 0.22% tensile-strained n-type Ge (indirect bandgap material) showed *decreasing* direct bandgap PL intensities at decreased temperatures¹⁷. In Ref. 17, this atypical PL behavior was explicitly attributed to the fact that 0.22% tensile-

strained n-Ge is *not* a direct-gap material; as temperature decreases and the Fermi function becomes more step-like, the carrier concentration at Γ decreases exponentially, resulting in quenching of the PL intensity from direct-gap transitions.

Figure R2. **a**, Intensity-dependent PL at 77 K showing linear increase of max and integrated intensity, **b**, Temperature-dependent PL at fixed laser pumping intensity showing blue-shift of peak with decreasing temperature, **c**, Integrated intensity vs temperature (K).

To address the reviewer’s comment, we added the following paragraph in the *Raman polar patterns and optical properties* section of our manuscript:

“We also performed temperature-dependent PL and low-temperature ($T= 77$ K) excitation-dependent PL on the 3.6% Ge/InAlAs nanocomposite (Supplementary Info). The integrated PL intensity increases and the PL peak blue-shifts as the temperature decreases, which is consistent with direct bandgap radiative recombination^{16, 18}. Excitation-dependent PL revealed that the PL intensities vary linearly with the excitation density. The inset of Fig. 4b shows that annealing up to 600 °C can improve the PL intensity by $\sim 2\times$ which may result from a reduced density of point defects in and around the Ge nanostructures¹⁹. Further study of optical properties such as time-resolved PL²⁰ and photoacoustic spectroscopy¹¹ will be necessary to confirm the band structure of the highly tensile-strained Ge/InAlAs nanocomposite”.

For the Supplementary Information, we have added the figure above and the paragraph below:

“We performed low-temperature ($T = 77$ K) excitation-dependent PL (Fig. S13a) and found that both the peak intensity and integrated intensity vary linearly with the excitation density; a linear dependence of integrated PL intensity on pump power is extremely common in studies of novel optoelectronic materials, including InN¹² and InGaAsN¹³. The sub-quadratic relationship may be due to exciton recombination in the Ge nanostructures¹⁴ or loss of carriers to the non-radiative trap states at the Ge and InAlAs interfaces¹⁶. The temperature-dependent PL measurements in Fig. S13b and c display an integrated PL intensity increase and a blue-shift of the PL peaks as temperature decreases. Note that 0.22% tensile-strained n-type Ge (an indirect bandgap material) showed decreasing direct bandgap PL intensities at decreased temperatures¹⁷”.

Comment #5. Also there are important missing references. For example, in the introduction, there is no reference to back up the statement “the cross-over from indirect to direct is predicted at 2% biaxial tension”.

Response: We apologize for neglecting that important reference and have added it in to the statement, “the cross-over from indirect to direct is predicted at 2% biaxial tension”.

Reviewer #3

First, we are grateful for the comment that “**the study is systematic, comprehensive, and novel, and I do believe that it should be published in *Nature Communications*.**”

Comment #1. In the best-case scenario, the Ge composition is 95%, as seems to be the highest composition 'alloyed' material from the EDS. Al is shown in APT, why not show side-by-side Ge, As, Al, In? There is possibly a great deal of doping compensation going on. With dominant As, an n-type carrier density in excess of $1E18/cm^3$ is reported. This of course would not fold back to the incorporated As in Ge from EDS which shows that As is actually a few % of the composition. This leads to the question of whether one should mention Ge nanowire at all or just discuss Ge/InAlAs composite everywhere on the manuscript? What is the lattice incorporation mechanism, are there any predicted crystal structures or can first principles predict a stable alloy or compound structure?

Response: The manuscript (Fig. 2c EDX linescans) shows the compositions of the Ge/InAlAs nanocomposites. We added APT images for Ge, In, Al, and As elements side-by-side in the Supplementary Information with the sentence below.

“Fig S3 shows APT images from a Ge/InAlAs nanocomposite grown at $0.2 \mu m/hr$ and $500 \text{ }^\circ C$, clearly showing a lack of In, Al, and As elements where Ge nanowires exist”.

Nomenclature of this novel material can be confusing, since we have Ge and InAlAs as components that are intermixed, while the overall properties emerge from the composite structure. For example, the carrier concentration in the low 10^{18} cm^{-3} range likely comes from a complex interplay of doping and compensation and is thus a property of the nanocomposite; we cannot really say what the carrier concentrations are in the separate components.

However, based on the composition studies by EDX linescans and APT, the Ge nanostructure component of the nanocomposite is >95% Ge at the center, while the InAlAs matrix is <<1% Ge. Thus, we feel that in discussing the microstructure, it is reasonable to reference both the components individually and the nanocomposite as a whole. Throughout the manuscript, we have strived to stay consistent with this style to minimize confusion.

The lattice incorporation mechanism of spontaneous phase separation is well described by both experiment^{3, 21} and simulation²², and briefly in our manuscript as well. Briefly, the major driving force for Ge adatoms to nucleate and grow Ge nanostructures is the mutual immiscibility between Ge and InAlAs. We believe that once a Ge-rich nucleus forms on the surface, it becomes a sink for later-arriving Ge adatoms, though further work such as *in situ* scanning tunneling microscopy would be needed to prove this, as was done for

ErSb nanostructures embedded in GaSb³. Once the surface is buried by a subsequent layer, the surface structures are effectively “frozen” in the bulk²². When surface kinetics are inadequate, the Ge dissolves into the InAlAs lattice, and no nanostructures form. We further proved the importance of surface kinetics in the formation of the Ge nanostructures with the post-annealing experiment, which showed no phase separation from the Ge/InAlAs metastable alloys.

We are not aware of any stable compounds that would form among Ge-In-Al-As. Binary phase diagrams show that Ge-In and Ge-Al are both eutectic systems without any intermediate phases^{23,24}, while Ge-As can form two line compounds- GeAs and GeAs₂²⁵. Thus, one of our primary concerns when we initiated this work was the unintentional formation of these unwanted phases. However, analysis by RHEED and XRD showed no unexpected diffraction. Most importantly, TEM diffraction patterns and high-resolution LAADF-STEM images showed that Ge nanostructures and InAlAs matrix possess diamond and zinc-blende lattices, respectively.

Comment #2. What is the nature of the band-offsets in the InAlAs/Ge system? Any possible rectification effects or is the conduction band-offset negligible?

Response: The band-offsets at the interfaces between Ge nanostructures and InAlAs matrix can be different depending on whether the interface is dominated by group-III or group-V atoms bonding with Ge²⁰. Pavarelli *et al.* reported that a Ge quantum well (QW) with In_{0.3}Ga_{0.7}As barrier possesses a type-I band-offset for the group-III (In and Ga)/Ge interface and a type-II band-offset for the group-V(As)/Ge interface.

In our Ge/InAlAs nanocomposite, we think that similar band-offsets may exist at the interfaces. The As/Ge interfaces form type-II band-offset with electrons confined in the Gamma valley of the Ge conduction band. The difference is the group-III/Ge bonding. Replacing Ga atoms with Al atoms will increase the conduction band minima of the InAlAs matrix, which leads to more confinement for electrons in the Ge nanostructures compared to the Ge QW/In_{0.3}Ga_{0.7}As barrier system. Also, the high tensile strain will push down the Gamma valley in the Ge conduction band. Therefore, we expect ample electron confinement in the conduction band for both types of interfaces.

Rectification at such interfaces is definitely possible, but such effects are likely masked at room temperature and were not studied in this work.

Comment #3. LED devices: 'well-behaved diodes with good rectification and low leakage current". Why not show currents on a log scale for leakage current visualization at least in supporting information?

Response: We agree with the reviewer’s suggestion. We added I-V curves in a semi-log scale in the Supplementary information with the sentence below.

“Fig. S12b shows I-V curves from the p-n nanocomposite LED and Ge-free InAlAs p-n junction in a semi-log plot”.

Comment #4. Some discussion on tensile stress in [100] orientation and its effects on different conduction band-edge minima would be helpful

Response: As shown by Chang *et al.*, biaxial tensile strain for growth along a [100] orientation is more favorable than [110] or [111] orientations for converting Ge into a direct-gap material, since it drops E_{g_Gamma} more strongly E_{g_L}²⁶. As briefly described in the *Strain properties of Ge/InAlAs nanocomposite* section of the manuscript and the Supplementary information, we speculate that Ge nanostructures far beneath the surface may take on a triaxial strain state. Yang *et al.* have calculated the effect of triaxial tensile strain in the band structures of Ge, and they have reported that only 0.7% isotropic tensile strain is required to convert Ge from indirect to direct bandgap material²⁷. Currently, we can only speculate on the

strain state of the Ge nanostructures far beneath the surface and will discuss the possibility of triaxial strain engineering in a future publication.

We added the sentences below to the *Strain analysis of Ge/InAlAs nanocomposite* section of the Supplementary Information to discuss tensile stress in the [100] orientation and its effects.

“Biaxial tensile strain for growth along a [100] orientation is theoretically predicted to be more favorable than [110] or [111] orientations for converting Ge into a direct-gap material, since it drops E_g more strongly than E_g .”

With these changes, we believe that we addressed all the questions and concerns raised by the reviewers. We look forward to hearing from you soon.

References

1. Moshnyaga V., *et al.* Structural phase transition at the percolation threshold in epitaxial $(La_{0.7}Ca_{0.3}MnO_3)_{1-x}:(MgO)_x$ nanocomposite films. *Nat. Mater.* **2**, 247-252 (2003).
2. Singer K. E., Rutter P., Peaker A. R., Wright A. C. Self-Organizing Growth of Erbium Arsenide Quantum Dots and Wires in Gallium-Arsenide by Molecular-Beam Epitaxy. *Appl. Phys. Lett.* **64**, 707-709 (1994).
3. Kawasaki J. K., Schultz B. D., Lu H., Gossard A. C., Pamstrom C. J. Surface-Mediated Tunable Self-Assembly of Single Crystal Semimetallic ErSb/GaSb Nanocomposite Structures. *Nano Lett.* **13**, 2895-2901 (2013).
4. MacManus-Driscoll J. L., *et al.* Strain control and spontaneous phase ordering in vertical nanocomposite heteroepitaxial thin films. *Nat. Mater.* **7**, 314-320 (2008).
5. Harrington S. A., *et al.* Thick lead-free ferroelectric films with high Curie temperatures through nanocomposite-induced strain. *Nat. Nanotechnol.* **6**, 491-495 (2011).
6. Norman A. G., *et al.* Ge-related faceting and segregation during the growth of metastable $(GaAs)_{1-x}(Ge_2)_x$ alloy layers by metal-organic vapor-phase epitaxy. *Appl. Phys. Lett.* **74**, 1382-1384 (1999).
7. Mukherjee S., Watanabe H., Isheim D., Seidman D. N., Moutanabbir O. Laser-Assisted Field Evaporation and Three-Dimensional Atom-by-Atom Mapping of Diamond Isotopic Homojunctions. *Nano. Lett.* **16**, 1335-1344 (2016).
8. Wang X., Shakouri A., Yu B., Sun X. H., Meyyappan M. Study of phonon modes in germanium nanowires. *J. Appl. Phys.* **102**, 014304 (2007).
9. Sood A. K., Menendez J., Cardona M., Ploog K. Resonance Raman-Scattering by Confined Lo and to Phonons in GaAs-AlAs Superlattices. *Phys. Rev. Lett.* **54**, 2111-2114 (1985).
10. Spitzer J., *et al.* Raman-Scattering by Optical Phonons in Isotopic $^{70}Ge_N^{74}Ge_N$ Superlattices. *Phys. Rev. Lett.* **72**, 1565-1568 (1994).
11. Eaves L., Vargas H., Williams P. J. Intrinsic and Deep-Level Photoacoustic-Spectroscopy of GaAs (Cr) and of Other Bulk Semiconductors. *Appl. Phys. Lett.* **38**, 768-770 (1981).

12. Wu J., *et al.* Unusual properties of the fundamental band gap of InN. *Appl. Phys. Lett.* **80**, 3967-3969 (2002).
13. Fehse R., *et al.* Evidence for large monomolecular recombination contribution to threshold current in 1.3 μm GaInNAs semiconductor lasers. *Electron. Lett.* **37**, 1518-1520 (2001).
14. Jin S. R., Zheng Y. L., Li A. Z. Characterization of photoluminescence intensity and efficiency of free excitons in semiconductor quantum well structures. *J. Appl. Phys.* **82**, 3870-3873 (1997).
15. Schubert M. F., *et al.* Effect of dislocation density on efficiency droop in GaInN/GaN light-emitting diodes. *Appl. Phys. Lett.* **91**, 231114 (2007).
16. Manna S., Katiyar A., Aluguri R., Ray S. K. Temperature dependent photoluminescence and electroluminescence characteristics of core-shell Ge-GeO₂ nanowires. *J. Phys. D: Appl. Phys.* **48**, 215103 (2015).
17. Sun X. C., Liu J. F., Kimerling L. C., Michel J. Direct gap photoluminescence of n-type tensile-strained Ge-on-Si. *Appl. Phys. Lett.* **95**, 011911 (2009).
18. Huo Y. J., *et al.* Strong enhancement of direct transition photoluminescence with highly tensile-strained Ge grown by molecular beam epitaxy. *Appl. Phys. Lett.* **98**, 011111 (2011).
19. Dashiell M. W., *et al.* Photoluminescence of ultrasmall Ge quantum dots grown by molecular-beam epitaxy at low temperatures. *Appl. Phys. Lett.* **80**, 1279-1281 (2002).
20. Pavarelli N., *et al.* Optical Emission of a Strained Direct-Band-Gap Ge Quantum Well Embedded Inside InGaAs Alloy Layers. *Phys. Rev. Lett.* **110**, 177404 (2013).
21. Tian Y., *et al.* Ultrahigh-Density sub-10 nm Nanowire Array Formation via Surface-Controlled Phase Separation. *Nano Lett.* **14**, 4328-4333 (2014).
22. Adams C. D., Srolovitz D. J., Atzmon M. Monte-Carlo Simulation of Phase-Separation during Thin-Film Codeposition. *J. Appl. Phys.* **74**, 1707-1715 (1993).
23. Olesinski R. W., *et al.* The Ge-In (Germanium-Indium) system. Bulletin of Alloy Phase Diagrams **6**, 536-539 (1985)
24. McAllister A. J., *et al.* The Al-Ge (Aluminum-Germanium) system. Bulletin of Alloy Phase Diagrams **5**, 341-347 (1984)
25. Olesinski R. W., *et al.* The As-Ge (Arsenic-Germanium) system. Bulletin of Alloy Phase Diagrams **6**, 250-254 (1985)
26. Chang G. E., Cheng H. H. Optical gain of germanium infrared lasers on different crystal orientations. *J. Phys. D: Appl. Phys.* **46**, 065103 (2013).
27. Yang C. H., *et al.* Dependence of electronic properties of germanium on the in-plane biaxial tensile strains. *Physica B* **427**, 62-67 (2013).

REVIEWERS' COMMENTS:

Reviewer #1 (Remarks to the Author):

The authors have adequately addressed the issues that were brought up in my first review. I recommend publication.

Reviewer #2 (Remarks to the Author):

Comment: Revised version of the manuscript with more detail on Raman study and temperature dependent PL improves the justification regarding the high strain observed in the nanowire and possible direct band gap transition. Specially, temperature and power dependent PL provides a good indication that the light emission could be of direct nature. Though the growth and structural study for nanocomposite is extensive I still think claim for record strain and study of band gap is preliminary. For example, as the authors are claiming a record number for strain, decoupling the phonon confinement contribution from Raman shift should be included to get the correct value for strain. In parallel, they should have included some other technique to support the high strain observed in the nanowires (as the high strain is the main selling point of the manuscript). Also I think authors should include some temperature dependent PL data in the main manuscript as this is important information for preliminary estimation of band structure. But I agree this preliminary study will most likely encourage further work by other groups (and perhaps this group) to get a better understanding of the qualitative and quantitative strain observed in these structures and the underlying photophysics of this nanocomposite. For this reason, I think the paper is probably suitable for publication.

Reviewer #3 (Remarks to the Author):

The authors have addressed the concerns I had and made changes in the manuscript to clarify. I believe that the manuscript can now be accepted for publication in Nature Communications.

Reviewer #1

We are grateful for the comment that “The authors have adequately addressed the issues that were brought up in my first review. I recommend publication”.

Reviewer #2

Comment: Revised version of the manuscript with more detail on Raman study and temperature dependent PL improves the justification regarding the high strain observed in the nanowire and possible direct band gap transition. Specially, temperature and power dependent PL provides a good indication that the light emission could be of direct nature. Though the growth and structural study for nanocomposite is extensive I still think claim for record strain and study of band gap is preliminary. For example, as the authors are claiming a record number for strain, decoupling the phonon confinement contribution from Raman shift should be included to get the correct value for strain. In parallel, they should have included some other technique to support the high strain observed in the nanowires (as the high strain is the main selling point of the manuscript). Also I think authors should include some temperature dependent PL data in the main manuscript as this is important information for preliminary estimation of band structure.

But I agree this preliminary study will most likely encourage further work by other groups (and perhaps this group) to get a better understanding of the qualitative and quantitative strain observed in these structures and the underlying photophysics of this nanocomposite. For this reason, I think the paper is probably suitable for publication.

Response: We are grateful that reviewer #2 agreed that our paper is suitable for Nature Communications. The reviewer suggested including some temperature dependent PL data in the main manuscript. We have already added a paragraph in the revised manuscript after the 1st round of review. More thorough strain analysis on the Ge/InAlAs nanocomposites such as decoupling possible phonon confinement is going to be conducted in the near future.

Reviewer #3

We are also grateful for the comment that “The authors have addressed the concerns I had and made changes in the manuscript to clarify. I believe that the manuscript can now be accepted for publication in Nature Communications”.

We believe that we addressed all the questions and concerns raised by the reviewers. We look forward to hearing from you soon.